# *Bacillus anthracis*, “la maladie du charbon”, Toxins, and Institut Pasteur

**DOI:** 10.3390/toxins16020066

**Published:** 2024-01-26

**Authors:** Pierre L. Goossens

**Affiliations:** Yersinia, Institut Pasteur, 28 rue du Dr Roux, 75015 Paris, France; pierre.goossens@pasteur.fr

**Keywords:** *Bacillus anthracis*, anthrax, *anthracis* toxins, Institut Pasteur, vaccines, regulations, societal control

## Abstract

Institut Pasteur and *Bacillus anthracis* have enjoyed a relationship lasting almost 120 years, starting from its foundation and the pioneering work of Louis Pasteur in the nascent fields of microbiology and vaccination, and blooming after 1986 following the molecular biology/genetic revolution. This contribution will give a historical overview of these two research eras, taking advantage of the archives conserved at Institut Pasteur. The first era mainly focused on the production, characterisation, surveillance and improvement of veterinary anthrax vaccines; the concepts and technologies with which to reach a deep understanding of this research field were not yet available. The second period saw a new era of *B. anthracis* research at Institut Pasteur, with the anthrax laboratory developing a multi-disciplinary approach, ranging from structural analysis, biochemistry, genetic expression, and regulation to bacterial-host cell interactions, *in vivo* pathogenicity, and therapy development; this led to the comprehensive unravelling of many facets of this toxi-infection. *B. anthracis* may exemplify some general points on how science is performed in a given society at a given time and how a scientific research domain evolves. A striking illustration can be seen in the additive layers of regulations that were implemented from the beginning of the 21st century and their impact on *B. anthracis* research. *B. anthracis* and anthrax are complex systems that raise many valuable questions regarding basic research. One may hope that *B. anthracis* research will be re-initiated under favourable circumstances later at Institut Pasteur.

## 1. Introduction

This special issue deals with the contributions of the scientists at Institut Pasteur in the field of toxins. This essay will essentially focus on work performed on *Bacillus anthracis*, its toxins, and “la maladie du charbon” (anthrax) at Institut Pasteur from its origins to the present time. From time to time, crucial contributions from outside Institut Pasteur will be mentioned, but not extensively, as this is not an exhaustive review of anthrax research. This will tend to be a historical essay, a personal point of view (not an opinion), that is sometimes subjective, as it gives my own perception as a scientist who lived through a crucial change in the way science is performed in our rapidly evolving society.

In the first part, this review will present the history of anthrax research at Institut Pasteur from the end of the 19th century to the 1970s, which mainly covers the development and production of the first anthrax vaccines.


*This section mainly relies on the archives kept at the Centre de ressources en information scientifique (CeRIS) at Institut Pasteur. As archival research is never completed, there is an opportunity for further studies in this domain (for editorial reasons, the footnotes have been inserted into the text; they can be read or skipped according to readers’ preference).*


The second part will concentrate on the 1986–2015 period, exemplifying the richness of original research approaches to microbiology, toxins, and therapeutics.

*B. anthracis*, the bacterium responsible for “la maladie du charbon” (in French) (Milzbrand in German and anthrax in English) has, from the very early stages, a common history with Louis Pasteur. *B. anthracis* was first observed by Rayer and Davaine in 1850 [1]. It then became the centre of an intense controversy between Louis Pasteur and Robert Koch. 

*Pasteur and Koch were engaged, at the time, in a fierce scientific competition in the nascent field of microbiology. Many factors interfered, leading to this clash of personalities: their age, scientific recognition, and the language barrier with unfortunate consequences ranging from a lack of knowledge of prior publications to deep misunderstandings (such as during the September 1882 Geneva congress), not forgetting the political context after the Franco-Prussian 1870 war. These details come from the highly informative book on the interactions between Pasteur and Koch, which is available in French [2]* *with a German translation [3]**; an English translation would be invaluable for the scientific community.*

Together, they proved that the bacterium was responsible for anthrax and that it could produce spores that account for the periodic resurgence of the disease in the so-called “cursed fields” (“champs maudits” in French [2,4]). On the basis of his work on anthrax and then later on tuberculosis, Koch later proposed his famous “postulates” that link a putative pathogen to a given infectious disease. At the time, anthrax made a strong impression through the first publicised bacterial vaccination by Louis Pasteur in 1881 at Pouilly le Fort (Figure 1) [5]. At the end of the 19th century, the world of bacteria was being discovered and in full expansion; microbes were shown to be the causal agents of numerous diseases.

Following the success and the immense interest aroused by the first human vaccination against rabies in July 1885. 

*Rabies was a frightening disease that struck people’s imagination. As a zoonosis transmissible to humans, it was an ideal research field for Louis Pasteur for the application of the notion of pathogen attenuation to vaccination. This research was developed in Pasteur’s laboratory, then in École Normale rue d’Ulm in Paris, from 1880 until the first human vaccinations of Joseph Meister and Jean-Baptiste Jupille [6]*.

Pasteur reported on 1 March 1886 at the Académie des Sciences that 350 people had been vaccinated with only one failure. The rooms at the École Normale were becoming too limited in space to accommodate the increasing number of patients [6,7]. A project for a centre for vaccination against rabies was proposed. This led to the foundation and inauguration of Institut Pasteur at its current location, rue Dutot (now rue du Docteur Roux), on 14 November 1888.

## 2. The First Golden Age of *B. anthracis* Research: The Vaccines

*B. anthracis* was known to be responsible for anthrax in domestic and wild herbivorous animals, causing a significant economic problem in the 19th century and the beginning of the 20th century, with massive livestock deaths, along with human infections, especially in wool sorters [4,8,9,10]. The discovery and the use of attenuated strains by Pasteur led to the introduction of effective vaccination, beginning with chicken cholera. The anthrax vaccine was a live attenuated vaccine first produced at École Normale and then at Institut Pasteur. 


*Pasteur initiated his research on B. anthracis in 1877 at École Normale. From the archives at Musée Pasteur, it appears that the “vaccin charbonneux” was produced as early as 1882 on premises rue Vauquelin close to the École Normale (AIP PAS. G1 46).*


A document from 1883 shows Louis Pasteur as a business manager (Figure 2) annotating each page of the expenditure statement:

“This statement shows that all expenses relating to the vaccine were paid from the vaccine fund. What remains in the fund forms a net profit which is divided into five parts: Mr. Pasteur reserves two parts for himself; he allocates two parts, i.e., an equal sum, to his collaborators. The fifth part constitutes a reserve fund”.

“The annual sum allocated to my laboratory by the *Ministère de l’Instruction Publique* is ten thousand francs. At the end of the year, this is barely enough to pay for gas and heating”.

“In 1882, 10,000 fr. of the 50,000 remain to be spent. This credit of 50,000 is just enough to cover the laboratory’s working expenses. 24 May 1883”.

At this time, vaccines relied on the empirical process of the attenuation of pathogens (as will be discussed later, the basis of the pathogenesis of *B. anthracis* was unknown—the toxins were only fully described in 1954 [11], and the plasmids carrying the genes coding for the toxins and capsule were reported in 1983 and 1985 [12,13,14]). The stability of the attenuation level was far from mastered, leading to the low, but existing, frequency of adverse effects in vaccinated animals (AIP SVV.1). Furthermore, as the vaccine was alive, it needed to be maintained through *in vitro* (rarely *in vivo*) passages and its attenuation controlled.

Historically (AIP SVV.1 & SVV.2), the anthrax vaccines were developed and produced in the “Service des Vaccins Vétérinaires”, initially by Charles Chamberland from 1889 to 1904; the vaccine against “Rouget du porc” (swine erysipelas, *Erysipelothrix rhusiopathiae*) was the other major bacterial vaccine produced during this period.

The anthrax vaccine was distributed in many countries; the production was relocated to produce and distribute the vaccine directly on site. An interesting document gives some insights into this aspect. In 1886, a contract was signed between Charles Chamberland and Henri Lefebvre de Sainte Marie—defined as “député, sous-directeur, laboratoire de M. Pasteur” and “ancien inspecteur général de l’agriculture”, respectively; it aimed to entrust the production of the anthrax vaccine outside France (and its colonies) for 30 years (Figure 3). Some countries were excluded, as other exclusivity contracts were already running; these were Argentina, Paraguay, Uruguay, and Austria-Hungary. The selling prices were defined. In another instance, a little later (1888–1891), Louis Pasteur sent Adrien Loir, his nephew, to Australia to confirm that Cumberland disease, which affected cattle on this continent, was, in fact, anthrax. Adrien Loir then set up a vaccine production unit in Sydney [15].

After Chamberland’s illness and death (1908), the “Service des Vaccins Vétérinaires” was headed by Émile Roux as “Chef de service” and Constant Jouan as “Chef de laboratoire adjoint”; Jouan actually managed the service, as Émile Roux had his own directorial activities. Jouan’s presence at Institut Pasteur can be traced back to 1893; he was “préparateur” in the “Service de Microbiologie appliquée à l’hygiène et des vaccinations”, managed by Chamberland. Then Jouan left the Institut Pasteur (before 1925).


*The exact date has not yet been found in the Archives at Institut Pasteur. Jouan then created the well-known laboratory equipment enterprise that still carries his name (see the following link for the 1933 catalogue: *
*
http://www.bium.univ-paris5.fr/histmed/medica/cote?extaphpin014
*
*, accessed on 5 November 2023). (All the information in this paragraph was researched and kindly communicated by Sandra Legout, CeRIS, Institut Pasteur)*


Victor Frasey, the director of the “écuries d’Alleray” (stables at Alleray, premises not far from the Institut Pasteur campus) then assumed responsibility, with Charles Truche and André Staub as “adjoints”. Truche later headed the service until 1934. André Staub was an “assistant” from October 1906 onwards; from 1934, he managed the “Service des Vaccins Vétérinaires” and maintained this activity until 1951, the date of his retirement (Figure 4).

When perusing the laboratory notebooks and the official reports of these structures, it emerges that the main research on *B. anthracis* focused on the production, characterisation, surveillance, and improvement of the vaccines. This type of approach was similar to that used for the vaccines against chicken cholera, swine erysipelas, or rabies.

Anthrax vaccination usually required two to three inoculations of the attenuated vaccine strains with increasing virulence, with the “first vaccine” being the most attenuated; thus, so-called first, second, and, sometimes, third vaccines were produced, as is apparent from the laboratory notebooks (AIP SVV.1&2). Their degree of residual virulence was regularly tested in mice, guinea pigs, and rabbits, resulting in the modification of their use. For instance in an entry on 15 September 1934 (Figure 5): “trial of the second anthrax vaccine (used as third vaccine from 22 September 1934) … trial of the third anthrax vaccine (used as second vaccine from 22 September 1934)” or “second anthrax vaccine … too strong, to be used as third vaccine (vials labelled 23 April 1940)” (translation PLG).

One of the particulars was that each animal species to be vaccinated showed different susceptibilities to anthrax: some are highly susceptible (sheep), others much less so (bovines); hence, the residual virulence of the less attenuated vaccines in susceptible species. Accidents of vaccination regularly occurred in the vaccinated animals, prompting additional research (AIP SVV.1&2). There were repeated attempts to obtain a “vaccin unique”, a single vaccine that could protect all animal species. However, this did not actually succeed, and *in fine* there were several “vaccin unique” that were either specific to sheep, bovines, or goats (“vaccin unique mouton”, “vaccin unique bovin”, and “vaccin unique chèvre”; AIP SVV.1&2). Usually, the bovine vaccine was a two-fold dose of the sheep vaccine.

In other cases, following instances of some vaccine batches having suspected low efficiency or in areas where there was a high level of *B. anthracis* spore contamination, the production of the vaccine strain was tailored by adapting its attenuation levels, with such denominations as “vaccin fort” (i.e., strong) and “vaccin special” found in the laboratory notebooks.

An intriguing point is what type of attenuation the Pasteur vaccines harboured, as the substratum of virulence was then unknown. Did they lose one of the plasmids, thus becoming atoxinogenic or unencapsulated? On 31 March 1922, André Staub explored this aspect (Figure 6), and the following is what he observed:

“First vaccine: around ¼ of encapsulated bacteria, normal capsule

Second vaccine: almost all bacteria are encapsulated, normal capsule, rare ‘amplified’ capsule

Third vaccine: almost all bacteria are encapsulated, very thick capsule” (my own translation).

Clearly, the “Pasteur” strains used were encapsulated but at different levels, either in terms of the percentage of the entire bacterial population or the quantity of capsular material per bacterium. They were most probably toxinogenic; testing toxinogenesis was, however, not available at this time, so this question will most probably remain unanswered. One hope would be to sequence the vaccine strains. Some (many?) strains have been labelled “Pasteur strains” in various laboratories throughout the world, originating from strain exchanges between laboratories. As they have been stored for many years, how much they reflect the original strains with the minimum of genetic changes during storage and cultivation remains to be evaluated. Some of these “Pasteur” strains were later tested for capsulation, showing capsulation heterogeneity and suggesting that some form of encapsulation reversal could occur [13,14]. Our current knowledge of genetics and gene regulation might provide a more specific basis for the phenotype of these strains (these approaches have tentatively been applied to some of the “Pasteur” strains [16,17]).

One last opportunity would be to exploit the ancient Pasteur first and second vaccine strains stored in sealed vials in the Institut Pasteur museum that have been preserved until the present day (Figure 7).


*The conservation of biological samples in a museum is an interesting topic, raising ethical concerns in terms of human sources and the regulatory concerns relevant to specific regulations (see Section 5*
*). Different approaches are usually considered, ranging from destruction—and the loss of biological patrimony— to storage and access according to regulations. Valorisation could be a key mission for a museum [18,19]*
*; in this respect, the collection in the Musée Pasteur might provide valuable data. Constant vigilance should, nevertheless, be exerted, following the evolution of the regulations and the consequences for biological patrimony through the degree of stringency of their application/implementation in each institution, Institut Pasteur included) to avoid irreversible decisions that might be regretted later.*


Due to the advances in sequencing and genetic analysis, their characterisation should bring interesting insights into this old question; one advantage is the fact that the vaccines exist in the form of spores that are highly resistant. Spores are hard to break, and it may prove challenging to extract enough DNA for meaningful sequencing in these limited, precious samples; they may also have accumulated some mutations due to cosmic radiation [20,21,22]. Let us be imaginative and optimistic.

In the 1930s, unencapsulated strains were reported by Nicolas Stamatin in Romania and Max Sterne in South Africa [23,24,25]. When used for vaccination, these strains gave lower mortality/morbidity and were safer for use on cattle [26]. At the time, nobody understood the basis of this attenuation, as the plasmids had not yet been described. The Sterne strain arrived at Institut Pasteur on 29 December 1947 through André Staub (Figure 8, AIP SVV.1) and was tested for its protective efficiency [27]. No report considering its potential use as an alternative vaccine was found in the laboratory notebooks.

Apart from the vaccines, immune sera were produced from rabbits and horses for diagnosis (“sérum précipitant anticharbonneux pour réaction d’Ascoli”) and for the experimental testing of protection transfer, with no great success achieved (AIP SVV.1).

When André Staub retired in 1951, the Service des Vaccins Vétérinaires was fused with the Service de Microbiologie Animale, which exerted its activity into research on viruses (an emerging domain at this time) under Henri Jacotot, André Vallée, and Bernard Virat as successive heads, assuring the permanence of *B. anthracis* activity until the beginning of the 1970s. In 1954, for instance, Bernard Virat assessed the longevity of *B. anthracis* spores from samples ranging from 1884 to 1900, gathered in the museum ([29] and AIP SMA.1 for the original data); only 4 out of 100 samples could be revived and three of them had kept their initial virulence but were unable to protect rabbits against a virulent challenge.

Looking back at the activity of anthrax vaccine production at the Institut Pasteur provides some hints as to the number of doses delivered. In 1914, for the 25th anniversary of the Institut Pasteur, Émile Roux mentioned the following:

“The oldest of our practical departments is that of the anthrax vaccine, it goes back to the famous experiment at Pouilly-le-Fort, in 1881, and was organised by Chamberland. Soon the vaccine for swine erysipelas was also developed and for the past thirty-two years, the department has delivered 41,649,592 doses of anthrax vaccine and 10,716,906 doses of swine erysipelas vaccine. Messrs. Jouan and Staub who ensure the preparation of these vaccines deserve the recognition of farmers” (english translation Dominique Goossens).


*“Le plus ancien de nos services pratiques est celui des vaccins charbonneux, il date de la célèbre expérience de Pouilly-le-Fort, en 1881, et fut organisé par Chamberland. Bientôt le vaccin du rouget des porcs vint s’ajouter à celui du charbon et depuis trente-deux ans que le service fonctionne, il a délivré 41 649 592 doses de vaccin charbonneux et 10 716 906 doses de vaccin du rouget. MM. Jouan et Staub, qui assurent la préparation de ces vaccins, ont droit à la reconnaissance des agriculteurs ” (discours de M le Docteur Roux, Le XXVe Anniversaire de l’Institut Pasteur. In: Revue internationale de l’enseignement, tome 67, Janvier–Juin 1914. pp. 60–82).*



*https://www.persee.fr/doc/revin_1775–6014_1914_num_67_1_6822 (accessed on 5 November 2023).*


In 1936, anthrax Pasteur vaccine production was in the range of 120,000 annual doses for sheep and 90,000 doses for bovine; in 1950, this rose to around 300,000 and 100,000 annual doses, respectively (Figure 9, AIP SMA.1).

In the 1960s, Institut Pasteur was experiencing a delicate financial period; during his directorship (1971–1976), Jacques Monod obtained an increase in financial support from the French government. Let us recall that, from its creation, Institut Pasteur was a private enterprise, in order to retain its independence. The downside was that it had an obligation to find financial sources, for example, through the industrial commercialisation of its products (vaccines, diagnostics, antisera, antitoxins, etc.). The acceptance of this public financial contribution was tied to the separation of the research activities and the production/diagnostic activities. This led to the creation of Institut Pasteur Production (1972–73), which was later split in two: Diagnostic Pasteur and Pasteur Vaccins. The latter was then fused within Institut Merieux where the Sterne vaccine was produced. The Pasteur anthrax vaccines, which were already in decline, then disappeared, and this was the end of an era [30].

It seems from the Archives that the vaccine was produced until the early 1970s when Bernard Virat deposited the *B. anthracis* strains in the collection of the Institut Pasteur. The activity around *B. anthracis* was quite low, mainly covering the functions of a current national reference center. Work on *B. anthracis* was then interrupted till 1986 (perusing the litterature shows that some experiments were performed in other laboratories, using *B. anthracis* as a tool/target for antimicrobial therapy assays (for instance, [31]).

## 3. Anthrax Toxins: The Puzzle of a Complex Research Domain

In the previous section, toxins produced by *B. anthracis* are not mentioned. This is not because they were not looked for. Other toxins were described as early as 1889–90 for diphtheria or tetanus, for example ([32,33,34,35], see this special issue [36,37]); toxins were even mentioned by Louis Pasteur for chicken cholera caused by *Pasteurella multocida* [38], though no confirmation could be obtained thereafter.

At Institut Pasteur, André Staub searched for the toxins produced by *B. anthracis* as soon as 1909, as can be read from his laboratory notebooks, and this carried on regularly going forward (for example in 1911 and 1920; AIP SVV.1&2). He did not, however, obtain clear and tangible results on toxic activities. Anne-Marie Staub, his daughter, and Pierre Grabar followed up this query later in the 1940s at Institut Pasteur, taking advantage of the progress in antibody purification techniques to explore which *B. anthracis* antigens could be involved in toxicity [39,40]; no clear demonstration could be achieved, the time was not ripe for such a breakthrough.

So why were so many years necessary to reach the basis of our current knowledge on *B. anthracis* toxins (initially acquired by Harry Smith from 1954 onwards, with a series of accompanying papers in the following years characterising the system)? Harry Smith has given an excellent, highly readable, and vivid account describing the conditions of this discovery [11].

Let us first begin with an *a posteriori* brief overview of what is currently known of *B. anthracis* toxins before turning to the potential reasons for this delayed description.

Two main toxic activities are produced by *B. anthracis*, which were considered (for many years) as (1) a lethal toxin leading to cellular death and (2) an edema toxin easily observed through the characteristic edema in anthrax or in animal models. However, these two activities are mediated through a third component, named protective antigen (PA), that ensures the entry of the catalytic components edema factor (EF) and lethal factor (LF) into the cell. PA multimerises—classically an heptamer—at the cell surface after interaction with a cell receptor. EF and LF then interact with two adjacent molecules of PA, with the heptamer thus accomodating three molecules of EF/LF (octamerisation has also been reported, thus accomodating four molecules of EF and/or LF [41]).

Historically, two toxins were described: edema and lethal toxins (ET and LT, respectively EF+PA and LF+PA). The current view is to name this complex a tripartite toxin, which can exert two toxic activities. After internalisation and intracellular trafficking (the complex events are out of the scope of this review; for extended notions, see specialised reviews, for example [42,43,44,45]), EF and LF are translocated into the cytosol, where they exert their enzymatic toxic activity.

In 1982, Steve Leppla showed that EF is a calmodulin-dependent adenylate cyclase [46]; LF enzymatic activity (zinc metalloproteinase) and cellular targets (the majority of mitogen-activated protein kinase kinases, MAPKKs) were unknown until 1994 and 1998, respectively [47,48,49].

Now, what could be the reasons for this lengthy delay before discovery when compared to other toxins such as diphtheria or tetanus toxin?

1. First, as just mentioned, the *B. anthracis* toxin(s) is a multi-component toxin, an AB toxin. Such multi-molecular architecture necessitates the purification of at least two components to be able to produce an active toxin that can be tested *in vitro* or *in vivo*. The detection of direct toxicity that mimics the pathology of the infection *per se* through the inoculation of filtered bacterial extracts or culture medium was, indeed, central to diphtheria or tetanus toxin discovery [32,33,34,35]. Furthermore, to follow the presence of a toxin in a given sample is much easier when its enzymatic activity is known (but not necessarily, as tetanus toxin enzymatic activity was discovered many years after its toxicity (1992 vs. 1889, see [50]). For *B. anthracis*, the edema was a pathognomical sign both in humans and animals; it could be easily followed *in vivo* during production and purification. However, *in vitro* experiments were hampered until calmodulin dependence was recognised [46].

2. Second, the production of the edema and lethal toxins by *B. anthracis* necessitates specific induction conditions [51] (toxin and capsule expression are co-ordinately regulated [52]). Many years were needed to finally unravel these *in vitro* conditions (bicarbonate + CO_2_), and thus mimic the *in vivo* environment to obtain sufficient levels of toxins. Harry Smith freed himself from these constraints and unknowns, as he purified the toxins directly from serum and peritoneal liquids from infected animals.

3. Third, another crucial aspect is the availability of biochemistry techniques used for the isolation and purification of biological molecules. The majority were developed and became available during the second half of the 20th century (electrophoresis, column purification, and immunotechniques, among others [53]) and could, thus, not be applied to in-depth analyses of the composition of the biological milieu (either procaryotic or eucaryotic). The main technique available at the turn of the 20th century was filtration (filtre de Chamberland).

Let us also keep in mind that the vaccines developed, initially at École Normale and later at Institut Pasteur, then internationally, were quite effective. When combined with the emergence of the live attenuated unencapsulated vaccines on the one hand [23,24], and the advent of antibiotics after the second world war on the other hand, veterinary anthrax was efficiently controlled. The economic pressure was less urgent, and this research domain was no longer a priority. However, as a consequence of the existence of programs for biological weaponry development, research was pursued in military/army laboratories, such as in Porton Down, UK, where Harry Smith encountered favourable conditions to develop his basic science project on *B. anthracis* toxins.

Taken together, this provides some clues as to why the *B. anthracis* toxins required a considerable amount of time before being unequivocally detected, purified, and characterised.

## 4. The Second Golden Age of *B. anthracis* Research

As mentioned above, the separation of the research activity from the diagnostics and vaccine activities at the beginning of the 1970s led to the interruption of research on *B. anthracis* at Institut Pasteur. Work on *B. anthracis* was revived from 1986 onwards at Institut Pasteur by Michèle Mock, a scientist at the CNRS, a national French research organisation (Centre National de la Recherche Scientifique). During her doctoral and post-doctoral formation, she specialised in colicins, which can be considered plasmid-borne bacterial toxins that are directed against other bacteria [54]. Her postdoctoral training in John Collier’s “toxin” laboratory (then in Los Angeles) made her fluent in the new techniques in molecular biology such as cloning and sequencing among many others [55]. The period was, indeed, blooming with new technologies in molecular biology. This was also applied to *B. anthracis*, hence the discovery of the genetic substratum of the toxins; the pXO1 “toxin plasmid” was first described in 1983 [12] and the “capsule plasmid” in 1985 [13,14], paving the way for future avenues of research.

Michèle Mock became aware of Steve Leppla’s work on *B. anthracis* edema factor and its calmodulin-dependent adenylate cyclase toxin activity [46] and decided to initiate a new route in her research career as she was keen to explore bacterial toxins; *B. anthracis* was now known to produce toxins. The scientific environment at the Institut Pasteur was favourable at that time. For instance, another bacterium that also produces a calmodulin-dependent adenylate cyclase, *Bordetella pertussis*, was actively studied in Agnès Ullmann’s laboratory on the floor below, and the scientific exchanges between the two laboratories were key to the successful emergence of the *B. anthracis* project. Furthermore the immense opportunity of the genetic tools developed by Patrick Trieu-Cuot that ultimately enabled the heterogramic transfer of genetic material between *Escherichia coli* and *B. anthracis* [56], allowed the generation of *B. anthracis* mutants. Everything was, thus, perfect for initiating a new era in *B. anthracis* research at Institut Pasteur, all the more as the senior scientists then in charge of the decisions gave their green light.

Interestingly, Michèle Mock’s *primum movens* was to understand the contribution of the *B. anthracis* virulence factors to the infection and, thus, explore the *in vivo* effect of inactivating each toxin—not only focusing on the genetics. Complementary expertise was, thus, introduced in the laboratory, leading to a multi-disciplinary approach, ranging from structural analysis, biochemistry, genetic expression and regulation, to bacterial-host cell interactions, *in vivo* pathogenicity in various animal models, and therapy development. The laboratory was a scientific hub for many colleagues, leading to scientific discussions, training, and the exchange of materials (almost impossible nowadays with our current regulations in France, see Section 5).

Summarising almost 25 years of *B. anthracis* original research from the anthrax Pasteur laboratory is quite a challenge. Some of the key points selected from this abundant research output are presented below:

### 4.1. On Toxins

If one summarises these years of research on the genetics of *B. anthracis*, the laboratory was a pioneer in terms of the construction of bacterial and plasmid systems for generating mutants, cloning the toxin genes (*cya*, *lef*, and *pagA* for the EF, LF, and PA moieties, respectively), and their subsequent inactivation through the insertion of antibiotic cassettes or point mutations [57,58,59,60].

In addition to exploring the contribution of each toxin component in virulence and pathogenicity, the inactivation of each gene enabled better purification of the remaining toxin for cellular or *in vivo* experiments, the aim being to purify them directly from *B. anthracis* and not from recombinant *E. coli*, as those toxins were contaminated with LPS, with its confounding multiple effects.

Similarly, another practical aim was to increase the production of toxins to increase purification yields; hence, the initial interest in regulation of toxin gene production. The regulation facet, which was, of course, also investigated for its fundamental scientific interest, was successfully developed by Agnès Fouet and her group, leading to the exploration of some of the central regulatory networks (*atxA*, *pagR*, and *codY*, among others [52,61,62,63,64,65]).

### 4.2. On Bacterial Cell Surface

The purified toxin components were initially contaminated with high molecular weight proteins from the vegetative cells; the *sap*/*eag* system of the S-layer, thus, became a research focus, both at the genetic (gene organisation, regulation) and structural level. Inactivating them was a means to allow better purification of the toxin components but, at the same time, opened the way to unravelling the regulation of the production of this surface layer of the vegetative cells [66,67,68,69].

Pursuing the studies on *B. anthracis* cell surface led to the exploration of the structure and regulation of the other main major virulence factor, the pseudoproteic poly-gamma-D-glutamate (PDGA) capsule [67,70,71,72]. Sortases and cell surface-anchored proteins also became a focus of interest [73,74].

In parallel to the exploration of the vegetative cell surface, the spore surface was extensively studied, as it was of interest for vaccine development; furthermore, spore surface composition and structure is central to sporulation and germination, hence its implication in successful colonisation during infection, this being an obvious target for vaccines [75,76].

### 4.3. On Pathophysiology

If one wishes to have a global view of the *B. anthracis* infectious process, the crucial point is to remember that anthrax is a toxi-infection with two facets: (1) one is the infection *per se*, i.e., the encapsulated bacteria disseminate systemically and multiply, leading to major terminal septicemia, and (2) the other is the deleterious effects of the toxins secreted by the multiplying bacteria on multiple organs and cellular systems (for this aspect, many excellent reviews are available, with just a few cited here [42,43,44,45]).

The studies at the anthrax Pasteur laboratory were initiated on the unencapsulated toxinogenic attenuated Sterne background [77,78,79], as it was easier and safer to manipulate. One of the drawbacks of this is that they reproduce only the toxin arm of the infection and necessitate specific animal models susceptible to toxin-only effects [80].

The exploration of the infection arm, i.e., infection with encapsulated non-toxinogenic strains, was developed in the laboratory from 2000 and developed by Pierre L. Goossens and his group [81,82]. Applying the bioluminescence technology to follow real-time *B. anthracis* infection *in vivo*, Ian J. Glomski deciphered the dynamics of bacterial dissemination from the portal of entry in cutaneous, inhalational, and gastric infections, either with encapsulated non-toxinogenic strains (exploring how the bacteria disseminate in the absence of toxins) or with toxinogenic unencapsulated strains (exploring how toxins interfere with the host defense mechanisms). The absence or presence of the poly-gamma-D-glutamate capsule strikingly modifies *B. anthracis* dissemination, pinpointing how crucial the animal model used could be to recapitulating anthrax [80,81,82,83,84].

A dual pattern of *in vivo* bacterial behaviour was unravelled during inhalational infection with the wild-type strain—both encapsulated and toxinogenic. Infection in each infected host progresses along two patterns of dissemination, either one mimicking what occurs when only ET (i.e., EF + PA) is expressed or another when only LT (i.e., LF + PA) is expressed [85]. This raises the intriguing possibility that the bacteria at the site of entry may initially preferentially express either ET or LT, with each toxin inducing different patterns of subsequent colonisation and dissemination. Is this a stochastic event at the bacterial level, or is this related to variations in the milieu that surrounds the bacteria at the portal of entry, thus influencing the ratio of EF/LF expression? A similar pattern of a temporal balance of EF/LF local secretion levels could be deduced from histological observations in the spleen (where depending on the size of the infectious foci—hence their “age”—an initial LF histological effect was followed by predominant edema provoked by EF) [85]. This describes a complex pattern of *B. anthracis in vivo* behaviour that will depend on local EF/LF production ratios and on the parameters in the local tissular micro-environment, such as the O_2_/CO_2_ balance (nasopharynx vs. lung, spleen, or liver) and temperature (cutaneous vs. deep organs).


*The influence of the local tissular micro-environment on pathogen-host interactions occurs in other diseases, such as cutaneous vs. visceral Leishmaniasis, in terms of temperature or Mycobacterium tuberculosis colonisation in different areas of the lungs, depending on the O_2_ tension: top vs. posterior areas for biped vs. quadruped behaviour (Gilles Marchal and Geneviève Milon, personal communication).*


Interestingly, although it is known that *B. anthracis* is a tripartite toxin, scientists in the anthrax field still reason as if there were two distinct toxins. As Mahtab Moayeri and Steve Leppla reflect: “The combinatorial toxins to this day remain named after the early observations made about their *in vivo* effects (lethality and edema)” [43]. Since the PA-heptamer binds three molecules of EF and/or LF, the ratio of EF/LF produced in the bacterial micro-environment will most probably influence the ratio of EF/LF molecules bound to PA, hence the quantity of each toxic moiety a cell is exposed to. Another question then emerges: if three molecules of EF and/or LF are bound to a PA-multimer, does each EF/LF moiety have the same probability of being translocated into the cytosol? In other terms, what is the probability of translocation for each remaining molecule? Is it equivalent for each, or is there a decrease of efficiency after each translocation event? The less favourable issue (for the bacterium) is that only one molecule can be translocated, implying that the PA-multimer would be trapping two potentially toxic molecules—an interesting and stimulating concept.

If one further pushes consideration on the cellular model, the complexity of the toxin effects increases; a single cell will bind various quantities of PA-multimers depending on cell surface receptor density, with each enabling the translocation of various quantities of EF and LF. As the intracellular pathways affected by each toxin are interdependent, synergistic or antagonistic effects will, in the end, lead to complex consequences for the cellular pathway functions, not forgetting that little is known about any possible disparity in *in vivo* EF vs. LF intra- and extra-cellular half-lives (extracellular proteases, proteasomes, etc.), hence the consequences on the intracellular EF/LF ratio and subsequent toxicity. All these points are still unclear and warrant further research.

### 4.4. On Therapeutics

When looking back at what anthrax represented in the 19th century, it was mainly a veterinary concern for the society of the time, particularly from an economic point of view; epidemics in livestock had dire consequences [8,9]. The antibiotic era had not yet begun, and the sole manner of control was knowledge of contaminated areas and carcass management. Antibiotherapy and vaccination drastically changed the philosophy of veterinary anthrax control; infected animals are usually disposed of or treated with antibiotics when needed, and vaccination protects the remaining livestock from any further extension of the epidemic. Research on veterinary anthrax is no longer a key research domain—veterinary viral infections are more deadly and pose more of a threat to global health.

Human anthrax usually develops from direct contact with infected animals or products [80]: cutaneous, through the manipulation of infected animals; inhalational—wool sorters’ disease (now, though rarely, through the resuspension of spores from contaminated skin used for making drums [86]); and digestive, through the ingestion of insufficiently cooked meat from infected animals. Human cases are extremely rare in developed countries due to adequate management. Anecdotally, an epidemic occurred in 2009 in drug users due to the injection of contaminated heroin [87]. There is no aerial human-to-human transmission, and basic protective measures (avoiding contact with potentially contaminated biological material) are usually sufficient. Whatever the origin, antibiotic therapy is the treatment of choice [88].

However, *B. anthracis* may have been used as a bioweapon for a long time from what is usually mentioned in the literature [4]. During the 20th century, programs in some countries have been carried out for such nefarious uses [89,90,91]. The anthrax letter events in the USA in 2001 [89,92] exemplified its potential effectiveness for malevolent purposes. The defence authorities in many countries, thus, increased their interest and support for anthrax research. Due to its expertise and central position both in France and internationally, the anthrax laboratory at Institut Pasteur had, indeed, been contacted in the 1990s for counselling to increase threat responsiveness; research programs were, thus, developed—therapeutic approaches in particular. They took advantage of the knowledge acquired throughout the years of *B. anthracis* research, expanding and suggesting novel avenues of basic research in return, with their potential future applications for disease control. This dialogue between applied and basic research was a major characteristic of the Institut Pasteur from its birth (and even earlier) when looking back on how Louis Pasteur developed his research axes, ranging from the tartrate studies to rabies vaccination [38,93].

These demands of the French defence authorities led to the establishment of a long-term collaboration with the IRBA (Institut de Recherche Biomédicale des Armées, the medical research structure of the army). The transfer of knowledge was central between the two research structures, and complementarity in the scientific approaches was key to this successful collaboration (for example, Jean-Nicolas Tournier was a pioneer of exploiting the high level technology of bi-photon imaging to follow in real time the dynamics of *B. anthracis* inhalational infection in the lung both *in vitro* and *in vivo* [94]), collaboration illustrated by a number of co-publications [95,96,97,98,99,100,101]. Similarly, collaborations were set up with our colleagues in the CEA (Commissariat à l’énergie atomique et aux énergies alternatives) for the development of rapid and sensitive detection technologies, either for toxins or spores [102,103,104,105].

#### 4.4.1. Preventive Therapies

The prevention of anthrax in humans relies on vaccination. The current human vaccines are based on PA being present in the bacterial culture supernatants of strains equivalent to the Sterne strain; these vaccines are produced in the US and UK. Their main aim is to target inhalational anthrax, should such an epidemic occur during warfare or bioterrorism action. These vaccines have answered the official requirements for vaccine development in humans.

However, as human anthrax is a rare disease, it means that one relies on experimental laboratory data and few epidemiological evaluations. Some concerns have been raised about the actual efficiency of the anthrax human vaccine [106]. Here emerges the problem of the choice of the animal model for testing therapeutics in general and, more specifically, for anthrax [80]. The animal model used will explore (to various degrees) each facet of the anthrax toxi-infection (see above). Thus, the PA-based vaccines relying on toxin neutralisation will mainly be tested in animals that are susceptible to the toxins. In contrast, animal models highly susceptible to the infection facet of anthrax will never be used for testing anti-toxin therapies, as the afforded protective effects would be masked by the overwhelming infectious process. In particular, the PA-based vaccines do not protect mice against an infection with a fully virulent wild-type strain, whatever the route of infection (and especially in inhalational anthrax, which is the most difficult infection to control).

The availability of a human vaccine in France could be problematic if urgently needed to protect given populations; hence, vaccination design for better protection in humans was also a main concern of the French MoD for the Pasteur anthrax laboratory, leading to the tentative development of a phase I human anthrax vaccine based on its experimental data.


*As will become apparent in Section 5*
*, the development of an anthrax human vaccine is subject to many hurdles; manipulating B. anthracis is not an easy task under current regulations. Drawing on our experience in such development, it seems judicious and reasonable to favour working on a strain belonging to the B. cereus group outside anthracis to produce spores, specific spore antigens, or the capsular poly-gamma-D-glutamate through the expression of its biosynthesis operon; unpublished experiments in the anthrax Pasteur laboratory found that B. cereus spores could replace B. anthracis spores, albeit with a small decrease in efficiency. Recombinant PA and LF as key vaccine components are already produced in E. coli. Due to the 500 bp rule of the French regulations governing any samples that may contain genetic material from B. anthracis (see below), such vaccine development concerning a human vaccine seems, for the time being, probably more easily (while safe) managed at the European level outside France.*


In order to summarise the almost 15-year vaccine project, the main aim was to target both arms of the infection through the addition of (i) formaldehyde-inactivated spores (FIS) and (ii) the poly-gamma-D-glutamate capsule naturally coupled to the peptidoglycan (PGC) to the PA-based vaccines. The end results saw full protection against subcutaneous infection and a high level of protection after an inhalational challenge with a fully virulent *B. anthracis* strain in the notoriously hard-to-protect mouse model ([57,70,95,107] and Fabien Dumetz, unpublished data, Figure 10A). Most interestingly, the anti-toxin humoral immune response could play a protective role even without toxin neutralisation; adding a toxin component to the spore immunisation led to a significant increase in protection against non-toxinogenic encapsulated strains (86.5% vs. 47.5%, Figure 10B; [57] and unpublished results). Both the humoral and CD4-T cell arms of the immune response could be involved by creating a micro-environment unfavourable to initial bacterial growth.

It is usually considered that humans are more sensitive to the toxemia (than to the infection) arm of anthrax and should be protected with the current PA-based vaccines [80]; so what was the rationale for this focus on the highly susceptible mouse model of anthrax? One key point is that experiments are usually performed on adult animals in the absence of infection, stress, or any other pathologies (except when stresses or age are the central aim of the study); the data are then translated in terms of humans in the same physiological state. The challenge is to ensure that the same level of protection is reached under a stress condition, be it physical, psychological, or in the presence of concurrent viral, bacterial, or parasitic infections. Anthrax vaccination interest relies, at least initially, on protecting defence personnel in the case of malevolent use as a biological weapon; the field conditions would most probably vastly differ from a normal physiological state. Will a human exposed to such stress conditions be “turned into a mouse” in terms of immune response and defence mechanisms, thus becoming highly susceptible to *B. anthracis* infection? A vaccine formula targeting both facets of the anthrax toxi-infection, thus, seems safer and warrants further scientific efforts. This is now apparent from the wider international interest in the use of capsular material as a co-vaccination antigen and in developing vaccines targeting spores, toxins, and bacillus [108,109].

#### 4.4.2. Curative Therapies

In addition to prevention through vaccination, curative therapies targeting the toxins or the bacterium were also studied in the Pasteur anthrax laboratory. Since their discovery, the *B. anthracis* toxins were considered the main virulence factors to be neutralised in anthrax. Monoclonal anti-PA, -LF and -EF antibodies were developed as soon as the technology was accessible [110]. These tools enabled the mapping of the domains of the toxin components, the quantitative detection and assay of the toxins, and the follow-up of their presence in experimental samples [77,111].

The therapeutic potential of inhibiting EF enzymatic activity with adefovir was shown through a collaboration with Wei-Jen Tang [85].

A collaboration with Gilles Guichard saw the proposal of a novel alternative to antimicrobial peptides (AMP), i.e., oligoureas, non-natural peptidomimetics that mimic the structure of AMP. They were as efficiently bactericidal as AMP, with the advantage of not being cleaved *in vivo*, thus increasing their half-life [112,113].

Another highly efficient molecule, both *in vitro* and *in vivo*, is secretory phospholipase A_2_-IIa (sPLA_2_-IIa), an anti-microbial agent well-studied in *Staphylococcus aureus*; it was anthracidal at the nano-molar level (collaboration Lhousseine Touqui [114,115,116]). Intriguingly, a correlation may exist between sPLA_2_-IIa levels and the degree of resistance to the infection arm of *B. anthracis* infection in different species. The human enzymatic molecule might, thus, be considered an interesting and highly valuable therapeutic in humans in some cases if needed. Anecdotally, we observed that allicin, the component mediating the pungent odour of garlic, was anthracidal (unpublished, following a query from Pr David Mirelman, who kindly provided the purified molecule [117,118]).

## 5. “MOT” (Special Agents) Regulation

*B. anthracis* is on the list of agents that can be used as a biological weapon. After the 2001 anthrax events in the USA [89,92], a flurry of regulations was implemented internationally. They aimed to regulate the biological activities in the laboratories, controlling the safety/security aspects of their use, including the dual-use notion, while keeping track of them and of the scientists manipulating them. Now, after more than 10 years of implementation, it seems worthwhile to evaluate how this radical change at the societal level has impacted research activities on *B. anthracis* (and all other concerned agents) over the longer term.

For its part, France, one of three countries with highly stringent regulations (with Canada and Singapore, as mentioned in a working document), decided on drastically restrictive regulations linked to judicial sanctions (MicroOrganismes et Toxines regulations, MOT).


*Links showing some of the evolutions of the French regulation (accessed on 5 November 2023):*



*2004: *
*
https://www.legifrance.gouv.fr/codes/id/LEGIARTI000006690180/2004-08-11/
*



*2012: *
*
https://www.legifrance.gouv.fr/codes/id/LEGIARTI000025104612/2012-05-01/
*



*2014: *
*
https://www.legifrance.gouv.fr/codes/id/LEGIARTI000028352295/2014-02-01/
*



*2023: *
*
https://www.legifrance.gouv.fr/loda/id/LEGIARTI000047600565/2023-05-29/
*


*and* *https://www.legifrance.gouv.fr/codes/section_lc/LEGITEXT000006072665/LEGISCTA000019217188/2023-08-04/*


*The sanctions in 2023 are up to 5–7 years’ imprisonment and a EUR 375,000–750,000 fine*


For example, one of the most restrictive rules was the “500 bp” rule, initially linked to the control of ricin gene manipulation but applied to all MOT items. Thus, any sample that may harbour genetic material of more than 500 bp in length originating from *B. anthracis* is considered MOT and should be declared, tracked, and stored in specific access-restricted locations.


*Historically, when these regulations were first implemented and their implication not yet mastered, any longer than 500 bp of DNA purified from B. anthracis was considered “MOT”, even if the same exact sequence could be found in another non-“MOT” micro-organism; similarly, any molecule purified from B. anthracis could be considered MOT, even if it was a molecule found in any living cells such as ATP. Fortunately, scientific soundness rapidly prevailed, and the regulations were more precisely specified.*


The MOT regulations led to the destruction of part of the biological patrimony in France—a study to approach this aspect would be of interest; any destructions were, of course, perfectly managed alongside all the safety concerns and procedures that already existed. Many laboratories and scientists did not wish to enter an indefinite, complex, and time-consuming circle of tracking and surveillance. Furthermore, from impromptu discussions, it emerged that scientists felt they were specifically targeted by this suspicion, and they questioned whether potential malevolent users would effectively be deterred. Let us not forget that *B. anthracis* is naturally present in France and in many countries, both at the endemic and epidemic level [88], and can easily enter our countries (refer to the inhalational and gastrointestinal anthrax cases originating from drum skins [86,119,120] among other reports).

The application of the regulations is sometimes so time- and mind-consuming, e.g., conforming to all procedures or waiting for the required official authorisations, that it can impede crucial scientific activity. For diagnosis, forensic, or therapeutic aims, the delay in receiving a strain or testing the efficacy of given therapeutics, such as vaccination, should be the shortest possible; strict adherence to regulation may have consequences, thus requiring a framework for specific emergency circumstances. This raises the points of preparedness in times of health urgency, societal or political pressure, and how to balance the need for effective control vs. efficiency in responsiveness to an emergency. Some positive evolutions have recently been observed, but more is needed to find alignment with the spirit of the intended aims of these necessary regulations, which is a titanic task (one may wonder whether the letter was followed more than the spirit of their intended aims, due to the potential judicial consequences).

Three additional layers of regulations were implemented from the beginning of the 21st century: animal ethics regulating animal use, safety regulations for BSL3 (biosafety level) laboratory and animal facilities, and regulations concerning genetically modified organisms. Research on *B. anthracis* is directly concerned with all counts. In some ways, all these regulations may interfere with the *bona fide* proposals of scientific projects and concepts, although illustrating the genuine and legitimate concerns of our current societies (from Galileo to Frankenstein and the sorcerer’s apprentice, scientists have been part of the collective imagination of society). It is delicate for us to think well or ill of the current situation, as we are intrinsically involved; it will depend on our future colleagues within their societies to evaluate the soundness of the way ours has decided to rein in our scientific activity (one may worry whether our expertise, preparedness, and reactivity could currently be made fragile if confronted with the use of *B. anthracis* as a biological weapon or, more generally, to any agent on the MOT list; let us remain optimistic).

## 6. Some Points of Discussion

Before concluding this review, some scientific considerations will be proposed to the thoughtful imagination of our colleagues in the anthrax field.

-First, let us stress one technical (although seeming trivial at first sight), in fact, a crucial aspect for a valid interpretation of the anthracidal effects of a given anti-bacterial agent, i.e., sporicidal vs. bactericidal. Usually, a certain incubation time for the tested molecule is required, during which germination may occur and the antibacterial molecule exerts its effect. Data interpretation should, thus, be drawn carefully, distinguishing between direct sporicidal activity and killing of germinated spores.-Through our studies, we observed that the cellular effects of the toxins were mediated *in fine* through epigenetic modifications of the intoxinated cell [121]. This opens the path of alternative therapeutics aimed at restoring the functionality of the intoxinated cells in reversing these epigenetic modifications. Such “resuscitation” therapeutics are not confined to anthrax and have been considered in other pathologies where epigenetic memory is thought to be part of the persistence of homeostasis perturbation [122].-Lung *B. anthracis* infection is a secondary infection from within capillaries, with the blood-circulating encapsulated bacteria being trapped due to their large diameter [123]. This phenomenon was already suggested at the beginning of the 20th century (cited in [124]). Conceptually, this may represent a more general way for a pathogen to reach the lungs; as a secondary event, this is an interesting concept to consider (for example, for encapsulated *Streptococcus pneumoniae* or *Yersinia pestis* at a given dissemination stage).-*B. anthracis* toxins and spores could be detected in the blood, liver, and/or spleen as early as 1 h after intra-nasal delivery [99]. Such rapid crossing of the respiratory epithelium might be a more general way for a pathogen to interact with its host. Protein delivery through the respiratory epithelium has, indeed, been well-studied for therapeutic means [125,126], though the exact mechanisms are still debated (through pores, between adjacent cells, transcytosis, etc.).-A crucial challenge is the induction of a respiratory immune response that will control inhalational anthrax. A recent report using BCG [127,128] raises the interesting concept that an immune response generated from within the lung capillaries could trigger a more effective protective immunity than from the aerial space. *B. anthracis* could be a valid candidate for such exploration.

## 7. Conclusions

In summary, until now, *Bacillus anthracis* and Institut Pasteur enjoyed almost 120 years, starting from the pioneering work of Louis Pasteur and his “lieutenants” in the nascent fields of microbiology and vaccination [2,6,38] and blooming between 1986 and 2010 following the molecular biology/genetic revolution (exemplified by the Nobel prize attributed to André Lwoff, Jacques Monod, and François Jacob at Institut Pasteur). The 1986–2010 anthrax laboratory was one of the first that was able to genetically manipulate *B. anthracis* and explore its many components—especially toxins—and their role in pathogenicity, leading to the comprehensive unravelling of many facets of anthrax toxi-infection. The tools and mutants developed in the laboratory were exchanged internationally and exploited in various laboratories before they were strictly controlled through the current regulations.

This second golden age of *B. anthracis* research is exemplified by the high-level scientific recognition and the many international collaborations and meeting participations that occurred; a special thought can be given to the International Conference on Anthrax beginning in 1989 in Winchester (UK), then which happened in 2003 in parallel with the third International Workshop on the molecular biology of *Bacillus cereus*, *Bacillus thuringiensis*, and *Bacillus anthracis* (expanded thereafter every two years as the BACT series, regrouping scientists from the *cereus*, *thuringiensis*, and *anthracis* fields, as these bacteria belong to the same group, i.e., *Bacillus cereus* [129] (Figure 11).

Indeed, in recent years, reports have emerged on *B. cereus* harbouring similar virulence factors as those found in *B. anthracis* (toxins and capsule), causing anthrax-like pathologies in humans and great apes [130,131,132]. In the course of the anthrax Euronet 2004–2006 European project [133], a highly stimulating collaboration on these *B. cereus* biovar *anthracis* from apes was developed between our colleagues from the Robert Koch Institute and our laboratory at Institut Pasteur, putting together our complementary and synergistic expertise [134]. A scientific collaboration was finally taking place between our two institutions more than 100 years after the fierce competition between Louis Pasteur and Robert Koch in the 1880s [2].

The *B. anthracis* case could serve to illustrate some general points on how science is performed in a given society at a given time and how a scientific research domain evolves.


*Many parameters obviously shape the way a given scientific activity can develop, such as the available technologies, the amount of investment in times of lack of financial resources, their mode of attribution/allocation, the degree of scientific openness and the way society aims to control it through specific regulations. The evolution of the global organisation and function of scientific structures in the 21st century led to a shortening of laboratory life duration and the advent of short-to-middle-term projects (typically two to three years).*



*All this aspects obviously influence the quality and orientation of research at the level of society. Moreover it questions their impact on the emergence of new concepts, technologies, or applications, as their basic and applied significance can be unpredictable, unexpected, and often unrealised until circumstances are suitable for their development.*


The 1990s and early 2000s were ideal for anthrax research at Institut Pasteur in Paris, capitalising on novel concepts, available technologies, scientific critical mass, positive scientific support, grants, and access to biological facilities for this pathogen, both at the laboratory and animal levels. The decision to terminate *B. anthracis* research at the Institut Pasteur was taken in 2015 at the institutional level. This raises the issue of a loss of knowledge and expertise in the anthrax field at the basic and applied levels in France.

*B. anthracis* and anthrax as a toxi-infection are complex systems that raise many valuable questions in basic research. One may hope that *B. anthracis* research will be initiated *de novo* later at Institut Pasteur with new technologies, mindsets, and ways of exploring the domain. In between, expertise will be lost and will need to be rediscovered through the data published by our predecessors; this is a challenging feat. Indeed, not all technical and experimental details are available in the published papers, regardless of the efforts and implications at the author, editor, and reviewer levels. A part that is missing is the transfer of tacit knowledge, wherein there always remain some unmentioned details [135].

Having experienced the *B. anthracis* field for many years, I noticed the many, often unwritten, hints given around in various anthrax laboratories, such as growth conditions or spore activation steps. The loss of laboratory memory is rapid, and information will need to be rediscovered (a waste of time, energy, and money, but at the same time, a great joy for the discoverer of a previously known and forgotten fact). Another missing part relates to the nature of language (even scientific) that plays a prominent role in this relative failure. On the one hand, an author cannot describe everything exhaustively.


*The French author Georges Perec explored this intriguing aspect in some of his works, such as “Tentative d’épuisement d’un lieu parisien” (An Attempt at Exhausting a Place in Paris), where the impossibility of describing absolutely everything was more than obvious. Michael Baxandall analysed similar attempts at describing pictures (Pattern of intentions, Introduction, Yale University Press) and reached the same conclusion.*


On the other hand, our language cannot describe everything; some blanks always remain [136].

Over these past years, regulation constraints have been implemented both at the national and institutional levels. Our young colleagues entering the anthrax field (or any MOT) experience a scientific world full of constraints (this could be called “the Matrix effect”, as they will live in a constrained world shaped by our society without sensing how it could be outside). On the positive side, such constraints might counterintuitively stimulate creativity.


*Georges Perec was keen to exploit constraints when stimulating creation in his works. Among the many playing with constraints, two are impressive: “la disparition” (translated as “a void”), around 300 pages without the vowel E, or “Les Revenentes” (translated as “The Exeter Text: Jewels, Secrets, Sex”) a novel of around 60 pages with only the vowel E, excluding all the others. In another artistic domain, Nadia Boulanger gave the same advice to jazzman Quincy Jones “Mieux il saura écrire avec des contraintes, plus il deviendra libre” (“the more he would know how to write with constraints, the freer he will be”;*



*https://www.radiofrance.fr/francemusique/podcasts/musicopolis/un-americain-a-paris-quincy-jones-entre-barclay-et-boulanger-1192593 (accessed on 5 November 2023)*


We can be confident in the ability of future scientists to take advantage of these constraints and be creative while keeping an open mind. As Robert Goossens wrote in 1957:

“The autonomy of life reveals itself unpredictably, the important point is to seize upon the apparently contradictory element of prevalent observation and to integrate it into knowledge instead of rejecting it as an aberration of nature […] since in experimental biology, the essential is not the single-minded pursuit of our initial aim, but the capacity of also seeing what we were not seeking”.

(*“L’autonomie de la vie se révèle imprévisiblement, le tout est de saisir le fait apparemment contradictoire à l’observation commune et de l’intégrer dans la connaissance au lieu de le rejeter comme une aberration de la nature […] car en expérimentation biologique, l’essentiel n’est pas de poursuivre toujours le but que l’on cherchait, mais de voir aussi ce qu’on n’y cherchait pas” p. 140*)[137]

Let us dream of a third golden age of *B. anthracis* research at Institut Pasteur.

## Figures and Tables

**Figure 1 toxins-16-00066-f001:**
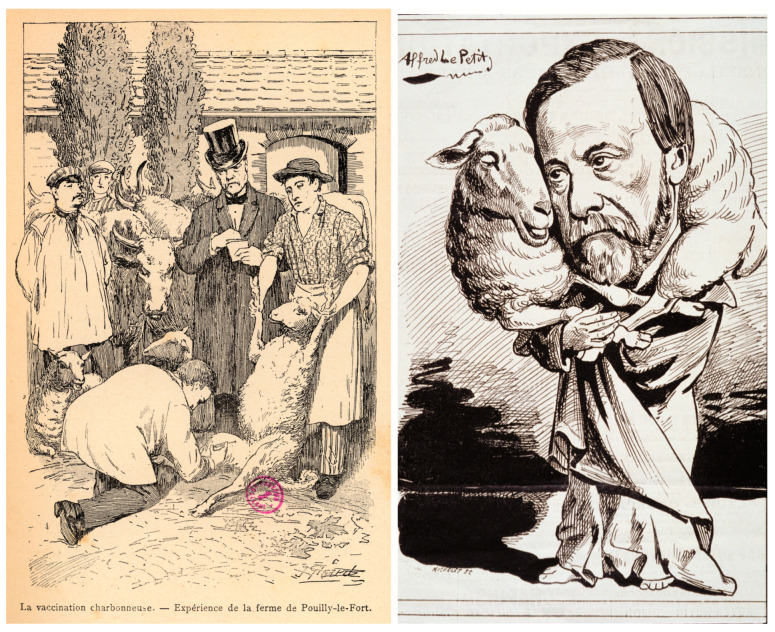
**First anthrax vaccination in 1881 at Pouilly le Fort**. (**left**) Emile Roux vaccinating sheep in the presence of Louis Pasteur. Drawing by J Girard, 1887. ©Institut Pasteur/Musée Pasteur. (**right**) Drawing by Alfred Le Petit in Le Charivari, 27 April 1882. https://gallica.bnf.fr/ark:/12148/bpt6k3073477c#, (accessed on 5 November 2023).

**Figure 2 toxins-16-00066-f002:**
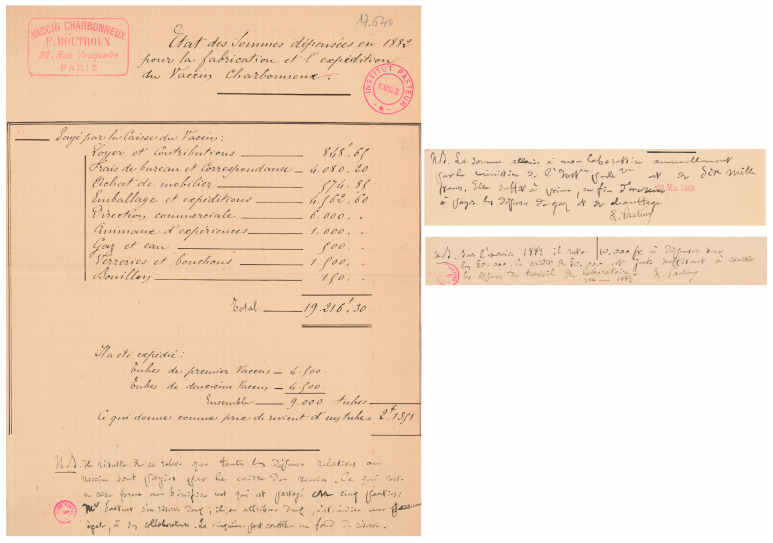
**Statement of manufacturing and shipping costs in 1882 for the “Vaccin Charbonneux”**. At the bottom of the page, Louis Pasteur’s autograph can be read; two others on two further pages of this document are shown on the right (english translation in the text). French transcription: “*Il résulte de ce relevé que toutes les dépenses relatives au vaccin sont payées par la caisse du vaccin. Ce qui reste en caisse forme un bénéfice net qui est partagé en cinq parties: Mr Pasteur s’en réserve deux; il en attribue deux, c’est-à-dire une somme égale, à ses collaborateurs. La cinquième part constitue un fonds de réserve”. “La somme allouée à mon laboratoire annuellement par le ministère de l’Instuction publique est de dix mille francs. Elle suffit à peine, en fin d’exercice, à payer les dépenses de gaz et de chauffage”. “Sur l’exercice 1882 il reste 10.000 fr. à dépenser sur les 50.000. Ce crédit de 50.000 est juste suffisant à couvrir les dépenses de travail du laboratoire. Le 24 mai 1883”*. AIP PAS. G1 46. ©Institut Pasteur/Musée Pasteur.

**Figure 3 toxins-16-00066-f003:**
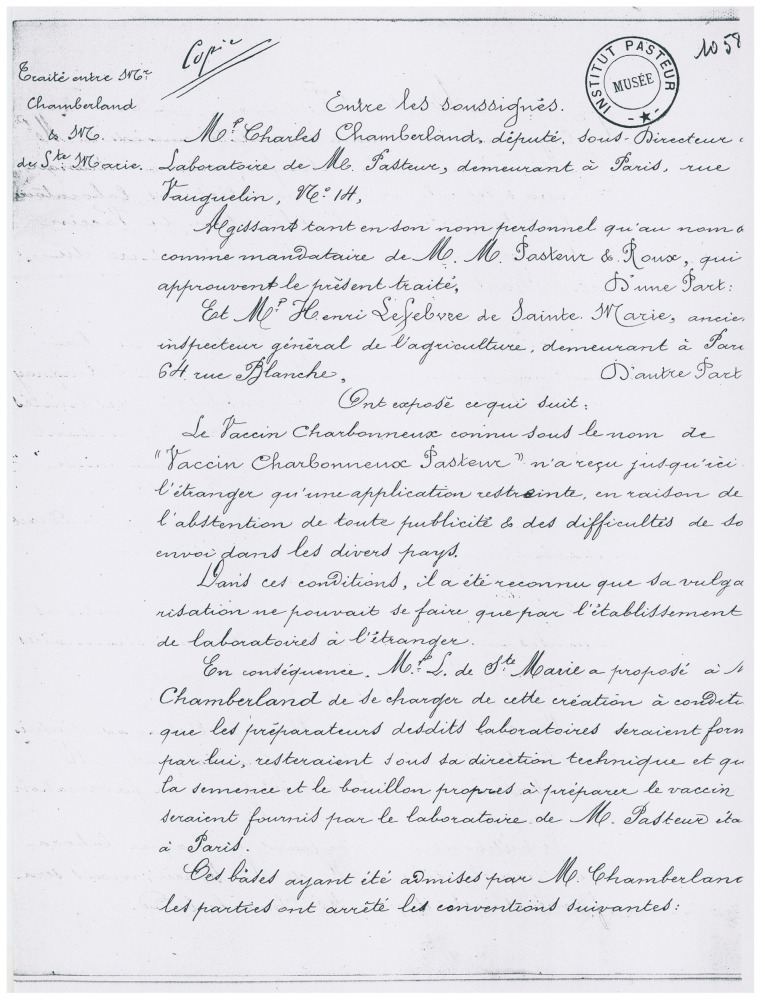
**Contract for the production of the anthrax vaccine in foreign countries (first page),** signed in 1886 between Charles Chamberland and Henri Lefebvre de Sainte Marie. Chamberland represented Pasteur, Roux, and himself: “The anthrax vaccine known as “Vaccin Charbonneux Pasteur” has so far received only limited application abroad, due to the absence of any publicity and the difficulties of shipping it to various countries. Under these conditions, it was recognized that its wider distribution could only be achieved through the establishment of laboratories abroad. Consequently, Mr. L. de Ste Marie proposed to Mr. Chamberland that he take charge of the creation of such laboratories, on condition that the assistants of the said laboratories, trained by him, would remain under his technical direction, and that the seed and broth suitable for preparing the vaccine would be supplied by Mr. Pasteur’s laboratory in Paris. These terms having been accepted by Mr. Chamberland, the parties entered into the following agreements:” (english translation: Dominique Goossens). “*Le vaccin charbonneux connu sous le nom de “Vaccin Charbonneux Pasteur” n’a reçu jusqu’ici à l’étranger qu’une application restreinte, en raison de l’abstention de toute publicité & des difficultés de son envoi dans les divers pays. Dans ces conditions, il a été reconnu que sa vulgarisation ne pouvait se faire que par l’établissement de laboratoires à l’étranger. En conséquence, Mr L. de Ste Marie a proposé à Mr Chamberland de se charger de cette création à condition que les préparateurs desdits laboratoires seraient formés par lui, resteraient sous sa direction technique et que la semence et le bouillon propres à préparer le vaccin seraient fournis par le laboratoire de M. Pasteur étant à Paris. Ces bases ayant été admises par M. Chamberland, les parties ont arrêté les conventions suivantes*:” AIP PAS.G1 33 ©Institut Pasteur/Musée Pasteur.

**Figure 4 toxins-16-00066-f004:**
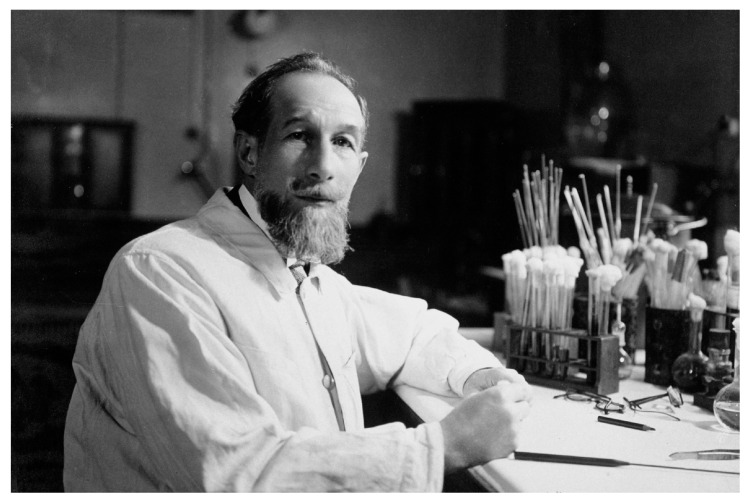
**André Staub (*circa* 1935) worked on *B. anthracis* from 1906 to 1951**. Other pathogens were also studied, such as swine erysipelas, avian influenza, chicken cholera, contagious bovine pleuropneumonia, fowl typhoid, and classical swine fever. His laboratory notebooks (spanning 1901–1951), which are kept in the Archives of Institut Pasteur, give a rare glimpse into the functioning of a research laboratory at the beginning of the 20th century. ©Institut Pasteur/Musée Pasteur.

**Figure 5 toxins-16-00066-f005:**
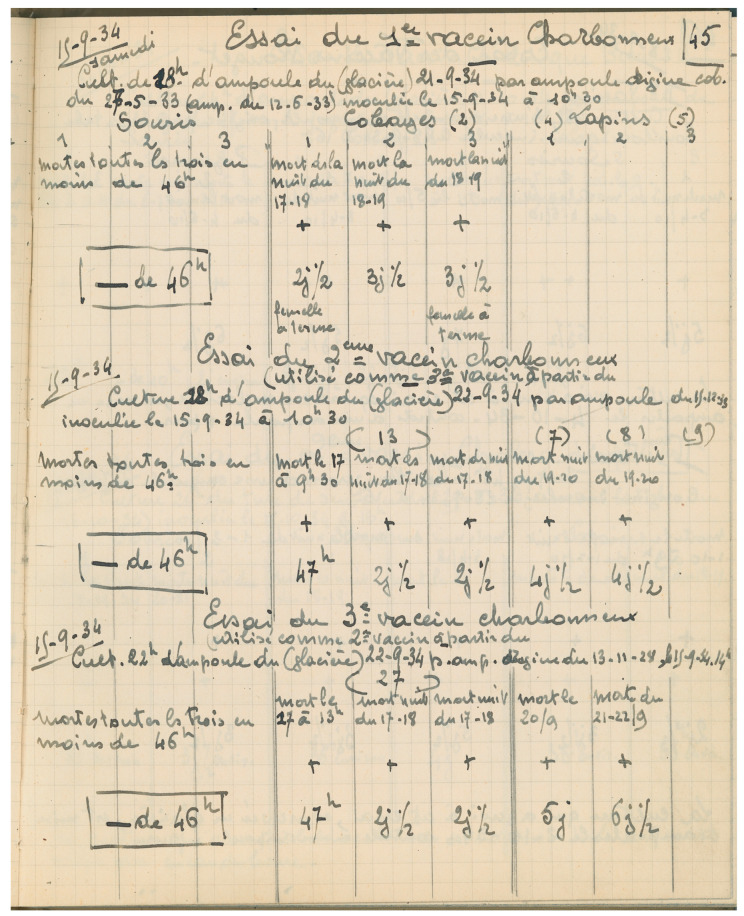
**Detail of André Staub’s laboratory notebook,** illustrating how, in 1934, the *B. anthracis* “Pasteur vaccines” were checked and adapted. ©Institut Pasteur/Archives—Fonds Service des vaccins vétérinaires.

**Figure 6 toxins-16-00066-f006:**
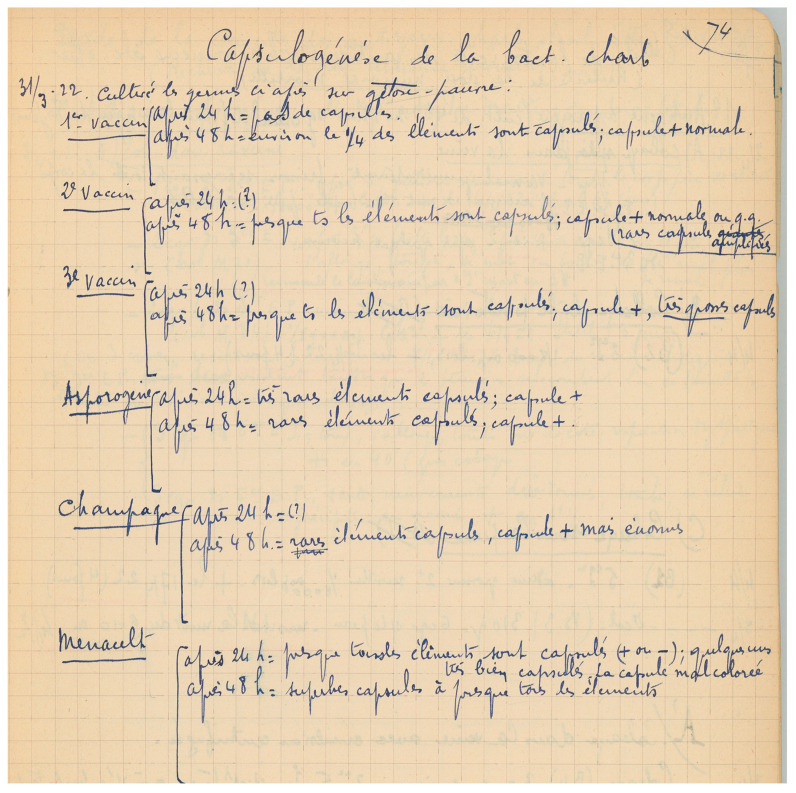
**In March 1922, André Staub tested the capsulation of the Pasteur vaccine strains.** The “asporogene” strain was related to assays developing another type of attenuated vaccinal strain; “Champagne” and “Menault” refer to two other strains that were addressed to the laboratory. ©Institut Pasteur/Archives—Fonds Service des vaccins vétérinaires.

**Figure 7 toxins-16-00066-f007:**
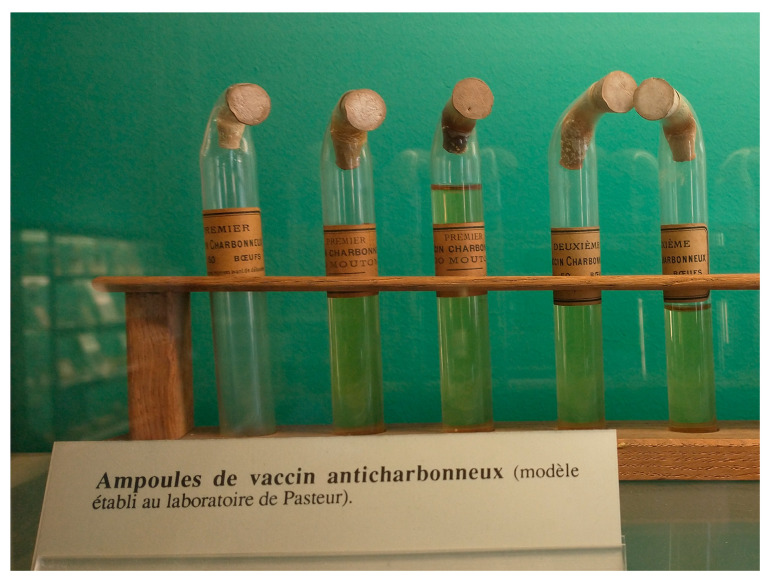
**Ancient vials of Pasteur vaccines form part of the museum collection;** exposed in the “salle des souvenirs scientifiques du Musée Pasteur”. Photo credit: Pierre L. Goossens.

**Figure 8 toxins-16-00066-f008:**
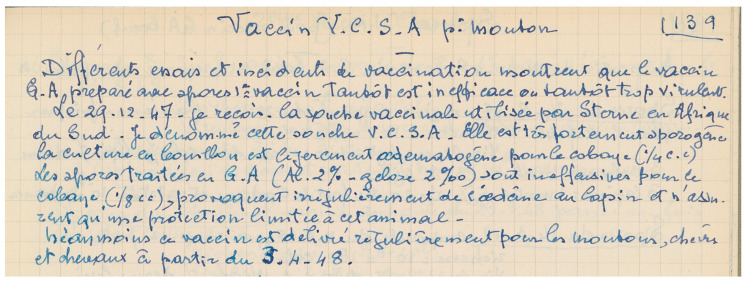
**The unencapsulated strain isolated by Max Sterne in South Africa reached the Institut Pasteur on 29 December 1947**, according to André Staub’s laboratory notebook entry. “Vaccine V.C.S.A. for sheep. Various trials and accidents of vaccination show that the G.A. vaccine prepared from spores of the first vaccine is sometimes inefficacious or sometimes too virulent. On 29 December 1947 I receive the vaccine strain used by Sterne in South Africa. I designate this strain V.C.S.A. It is highly sporogenic, the culture in broth medium is slightly edematogen for the guinea pig (1/4 cm^3^). The spores treated in G.A. (Al.2%–gelose 2‰) are innocuous for the guinea pig (1/8 cm^3^), irregularly provoke edema in rabbit and provide only a limited protection for this animal. However this vaccine is regularly delivered for sheep, goat and horses from 3 April 1948” (my own translation). G = gélose; A = Alun [28]. The meaning of the acronym V.C.S.A. is unknown, but may be guessed as Vaccin Charbonneux South Africa, awaiting further findings. ©Institut Pasteur/Archives—Fonds Service des vaccins vétérinaires.

**Figure 9 toxins-16-00066-f009:**
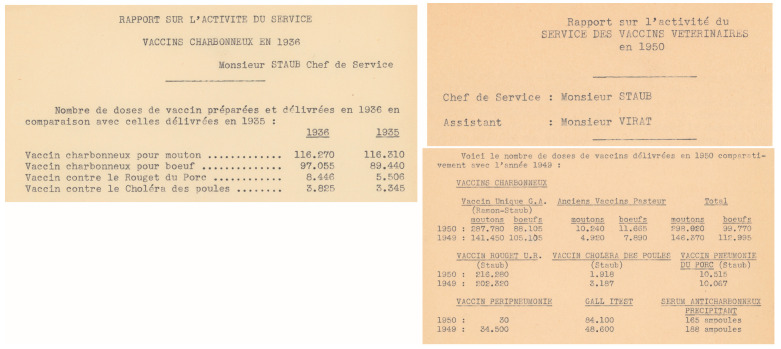
**Details from annual reports on vaccine production in the Service des vaccins vétérinaires** in 1935–36, and 1949–50, showing the number of doses of the various anthrax vaccines produced at Institut Pasteur, along with the other manufactured vaccines against animal pathogens. ©Institut Pasteur/Archives—Fonds Service des vaccins vétérinaires.

**Figure 10 toxins-16-00066-f010:**
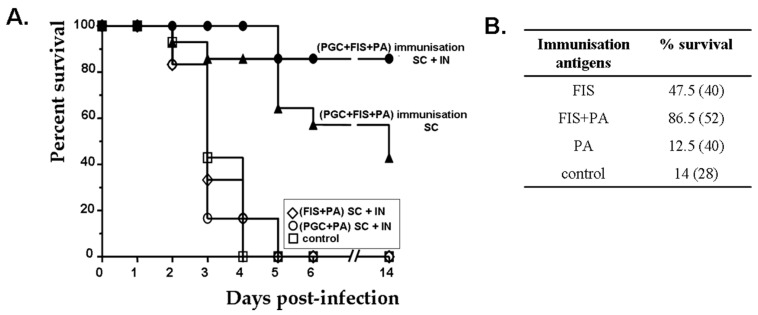
(**A**) **Significant protection against *B. anthracis* inhalational infection in the hard-to-protect mouse model.** Capsular poly-gamma-D-glutamate naturally anchored to the peptidoglycan (PGC) combined with formaldehyde-inactivated spores (FIS) and protective antigen (PA) were used as immunogens; double immunisation—systemic (SC) and mucosal (IN)—increased the afforded level of protection compared to single SC immunisation. Notably, the FIS + PA vaccine (as initially developed in [57] that fully protects against a cutaneous challenge) did not afford protection against inhalational infection, even after a combined SC + IN immunisation regimen. Immunisation protocols were as per [57] for FIS + PA, [70] for PGC, and [95] for the intranasal immunisation; the inhalational challenge dose of the fully virulent wild-type *B. anthracis* 9602 strain was 5.84 ± 0.10 log_10_ spores for outbred OF1 mice. The results were synthesised from four independent experiments; unpublished data from the postdoctoral work of Fabien Dumetz. (**B**) **The protection provided by anti-PA immunisation is mediated through neutralisation-independent mechanisms.** The immune response directed against the PA toxin moiety (FIS + PA) increases the protection afforded by spore-only (FIS) immunisation against a challenge with the non-toxinogenic encapsulated *B. anthracis* 9602P strain. The protocols of immunisation and challenge of OF1 mice were as per [57]. The results were synthesised from five independent experiments, with the number of animals in brackets.

**Figure 11 toxins-16-00066-f011:**
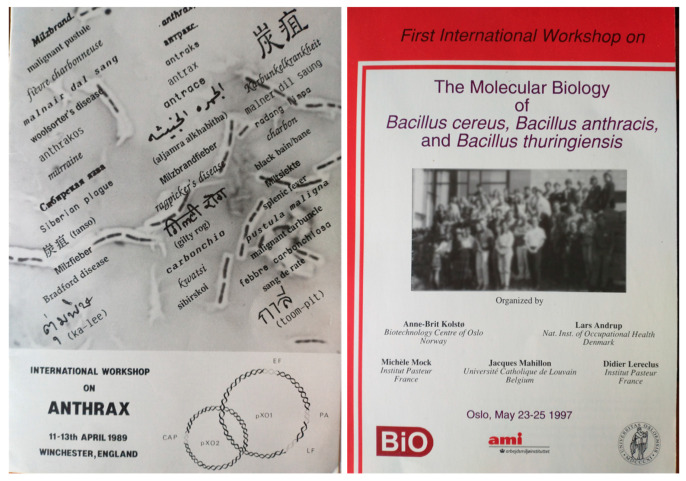
**Cover page of the two first international congresses** on anthrax in 1989 (Winchester, UK) and on the molecular biology of *the B. cereus* group in 1997 (Oslo, Norway). ©Michèle Mock, private archives.

## Data Availability

All documents of this historical review are available in the Archives of Institut Pasteur.

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
