# Peer review of "Bacillus anthracis, “la maladie du charbon”, Toxins, and Institut Pasteur"

_toxins, 2024, doi:10.3390/toxins16020066_

Round 1

Reviewer 1 Report

Comments and Suggestions for Authors

This review describes the history of Bacillus anthracis research at the Institut Pasteur from the late 1800’s and beyond. The review summarizes the early work on vaccines, the later work on toxins and the myriad of complexities involved in understanding their impact on infection. There is a rich history, and the figures supporting it were very interesting. The author concludes with a summary of the research conducted at the Institute in multiple areas and how this work is no longer being supported. This is indeed a very big loss. This review also provides an intriguing look at what areas might need focus in the future for anthrax. Nice discussion!

The concept of this review is very important and deserves the representation it will receive with this review. It is scientifically sound – with no experimentation and only discussion of prior work. Primarily, the writing needs some work. With the first reading, it was difficult to follow the different historical institutions involved in the research and the ‘Services’ of the Institut Pasteur, the changes, people, focus, and timelines. Some sentences were difficult to follow, and some needed lead in information for context before the statements. Often there is missing context to carryover ideas from one paragraph to another and sudden jumps to a new area with insufficient or no lead in statements. A few sentences were very long. With second review, some reading between the lines was still required – figuring things out rather than reading smoothly. Third reading allowed more appreciation of the content. Other suggestions listed below are related to specific areas where revisions in writing would be helpful. As a final note, this review is really important and hopefully, there is a way to focus a little more on the impact, the importance of the work and what it means to lose the expertise and collaborative potential of the experts at Institut Pasteur. Although it is stated at the end, it is low key. That is understandable for the authors. But if it is appropriate and possible, the authors might request a foreword from a significant collaborator(s) for this purpose. Just a thought.

Specific recommendations to improve the text and content:

1.       Overall, a table would be helpful with a timeline that includes at a minimum, ‘when’ for years covered, ‘which’ for the institute or branch of Institut Pasteur, ‘who’ for the people involved, and ‘what’ for the work done, as well as changes along the way, and periods when work was suspended. This would really support the text and ease the reader through the history.

2.       Give a hint in the first paragraph of the introduction what will be covered – specifically, the main sections, vaccines, toxins, and the research. This connects the reader through the following sections.

3.       Lines 41-42. The ‘intense controversy’ between Pasteur and Koch piques the readers interest but it’s not clear what the controversy is about. Readers can guess what it’s about but shouldn’t have to. The following sentence on lines 42-43 sort of implies what the controversy was about but that’s after the fact and not really clear. Explain the controversy. That will be more interesting too.

4.       Lines 44-48. The timeline jumps around a little with 1884 followed by 1881. Recommend moving the sentence about Koch postulates (1884) after the Louis Pasteur vaccination (1881). ‘Finally’ sounds like it occurred last – so it doesn’t really belong before vaccination – even though it may have been the most important piece at the time. They are all important pieces but a sequential timeline is easier to follow.

5.       Lines 56-62. As a pivotal time period leading to the foundation of Institut Pasteur, this deserves a little more context to lead into this section. First mention the rabies outbreak, how bad it was, the first rabies vaccine and its implementation. That makes it easier to appreciate the importance of the events and how they triggered growth in the area of vaccines, culminating in the Institut Pasteur.

6.       Lines 67-71. Timeline clarity. So, the Institut Pasteur was founded in 1888. This section mentions Ecole normale (include years) then Institut Pasteur. With the Pasteur document from 1883. Was this document from earlier work at Ecole normale? Include where the document is from so readers understand how it fits in the timeline.

7.       Lines 93-96. Long sentence. Very long stretch in parentheses. This needs to be reworked. Also, although this sentence is talking about ‘vaccines’, and most readers will know that attenuation is a standard and historic approach for ‘vaccines’, it would help someone unfamiliar to see the word vaccine in the first sentence. Maybe end the sentence after ‘empirical process.’

8.       Lines 126-129. This paragraph (actually one sentence) might belong with the prior paragraph. It is a later example of these activities where production techniques were shared in other countries.

9.       Lines 130-140.  Does the whole paragraph refer to one ‘service’? Readers need a reminder of which ‘service’ this refers to. Is it the Service des Vaccins Veterinaires? These changes can also be included in a table for support.

10.    Line 147 – structures. Does that mean laboratories? Associated with Institut Pasteur ‘service areas’?

11.    Lines 202-203. Genetic characterization and sequencing were done. This statement is not linked in anyway to prior discussion of heterogeneity in capsulation. Were any meaningful results found with the genetic characterization that can tie this statement to the prior discussion? Recommend adding something that makes this sentence meaningful.

12.    Lines 241-243. Longevity of spores. This sounds interesting but it would be even more so if the findings were included. What did they find? How many years or decades were spores still viable?

13.    Lines 272-276. This seems to suggest there was an interruption in B. anthracis research from the early 1970’s until 1986. Yet the example mentioned as evidence that some experiments were performed (ref 25) is from before this time period, 1966. This doesn’t fit the timeline. Was work on the vaccines continued until 1970’s but ‘research’ was interrupted before the 1970’s? Please clarify.

14.    Lines 327-328. Dependence of calmodulin for EF needs a reference.

15.    Lines 516-518. Place the parentheses explaining the acronym IRBA right after IRBA, then the medical research structure of the army at the end of the sentence.

16.    Lines 552-556. Interesting results. Discussion or thoughts on why adding toxin component to spores for vaccination gives better protection against non-toxigenic strain than spores alone?

17.    Line 629. Not familiar with the term ‘patrimony’ in this context. Is there an alternative to explain what was destroyed? Historical experimental collections?

Comments on the Quality of English Language

This needs some work. It is recommended that a native English speaker with good writing skills review the draft for sentence structure and flow. There are just a few minor instances listed below. 

In a few places, 'as soon as' for work of an early date should be replaced with 'as early as'. This means 'as early in time as'. 'Soon' is generally used to describe something that might be done as soon as possible or in the near future.

Line 179. Use 'lose' instead of 'loose'.

Footnote #12. Vaccine development is either 1) 'subject to' or 2) 'confronted by' many hurdles.

Lines 689. It's probably better to say 'genetically manipulate' than 'manipulate genetically.

Author Response

Referee 1:

This review describes the history of Bacillus anthracis research at the Institut Pasteur from the late 1800’s and beyond. The review summarizes the early work on vaccines, the later work on toxins and the myriad of complexities involved in understanding their impact on infection. There is a rich history, and the figures supporting it were very interesting. The author concludes with a summary of the research conducted at the Institute in multiple areas and how this work is no longer being supported. This is indeed a very big loss. This review also provides an intriguing look at what areas might need focus in the future for anthrax. Nice discussion!

The concept of this review is very important and deserves the representation it will receive with this review. It is scientifically sound – with no experimentation and only discussion of prior work. Primarily, the writing needs some work. With the first reading, it was difficult to follow the different historical institutions involved in the research and the ‘Services’ of the Institut Pasteur, the changes, people, focus, and timelines. Some sentences were difficult to follow, and some needed lead in information for context before the statements. Often there is missing context to carryover ideas from one paragraph to another and sudden jumps to a new area with insufficient or no lead in statements. A few sentences were very long. With second review, some reading between the lines was still required – figuring things out rather than reading smoothly. Third reading allowed more appreciation of the content. Other suggestions listed below are related to specific areas where revisions in writing would be helpful. As a final note, this review is really important and hopefully, there is a way to focus a little more on the impact, the importance of the work and what it means to lose the expertise and collaborative potential of the experts at Institut Pasteur. Although it is stated at the end, it is low key. That is understandable for the authors.

But if it is appropriate and possible, the authors might request a foreword from a significant collaborator(s) for this purpose. Just a thought.

We are fully in agreement with this suggestion. We did the utmost to remain the most factual possible and avoid any polemics. We would greatly appreciate such a possibility. Even part of the general comments of reviewers' 1 & 3 could cover this point. We will address this point with the guest editors of this special issue. We would appreciate  any suggestions from the Editor.

Specific recommendations to improve the text and content:

  1. Overall, a table would be helpful with a timeline that includes at a minimum, ‘when’ for years covered, ‘which’ for the institute or branch of Institut Pasteur, ‘who’ for the people involved, and ‘what’ for the work done, as well as changes along the way, and periods when work was suspended. This would really support the text and ease the reader through the history.

We agree that such illustration could be helpful. However, more research in the archives at Institut Pasteur will be needed as the denominations of the laboratories vary depending on the documents that are still available in different sections of the archives, the data for each scientist are sometimes scarce and need cross sectioning. For the time being we preferred to provide information one could be certain of, but will keep in mind this suggestion for future research.

  1. Give a hint in the first paragraph of the introduction what will be covered – specifically, the main sections, vaccines, toxins, and the research. This connects the reader through the following sections.

This is now added

  1. Lines 41-42. The ‘intense controversy’ between Pasteur and Koch piques the readers interest but it’s not clear what the controversy is about. Readers can guess what it’s about but shouldn’t have to. The following sentence on lines 42-43 sort of implies what the controversy was about but that’s after the fact and not really clear. Explain the controversy. That will be more interesting too.

This is now added

  1. Lines 44-48. The timeline jumps around a little with 1884 followed by 1881. Recommend moving the sentence about Koch postulates (1884) after the Louis Pasteur vaccination (1881). ‘Finally’ sounds like it occurred last – so it doesn’t really belong before vaccination – even though it may have been the most important piece at the time. They are all important pieces but a sequential timeline is easier to follow.

This has been modified accordingly

  1. Lines 56-62. As a pivotal time period leading to the foundation of Institut Pasteur, this deserves a little more context to lead into this section. First mention the rabies outbreak, how bad it was, the first rabies vaccine and its implementation. That makes it easier to appreciate the importance of the events and how they triggered growth in the area of vaccines, culminating in the Institut Pasteur.

This is now added

  1. Lines 67-71. Timeline clarity. So, the Institut Pasteur was founded in 1888. This section mentions Ecole normale (include years) then Institut Pasteur. With the Pasteur document from 1883. Was this document from earlier work at Ecole normale? Include where the document is from so readers understand how it fits in the timeline.

This has been addressed

  1. Lines 93-96. Long sentence. Very long stretch in parentheses. This needs to be reworked. Also, although this sentence is talking about ‘vaccines’, and most readers will know that attenuation is a standard and historic approach for ‘vaccines’, it would help someone unfamiliar to see the word vaccine in the first sentence. Maybe end the sentence after ‘empirical process.’

This has been reworked

  1. Lines 126-129. This paragraph (actually one sentence) might belong with the prior paragraph. It is a later example of these activities where production techniques were shared in other countries.

This has been modified accordingly

  1. Lines 130-140. Does the whole paragraph refer to one ‘service’? Readers need a reminder of which ‘service’ this refers to. Is it the Service des Vaccins Veterinaires? These changes can also be included in a table for support.

This addresses a frustrating point, as the denomination might vary according the documents and the persons writing them. This paragraph refers to the "Service des Vaccins Vétérinaires" as we were able to ascertain this point.

  1. Line 147 – structures. Does that mean laboratories? Associated with Institut Pasteur ‘service areas’?

We used the term "structure" to leave some uncertainty on the specific denomination at the time and avoid the anachronistic bias of projecting our own current definition to a past organisation. It is clear that further research would be interesting to understand the functioning mode of Institut Pasteur at the time.

  1. Lines 202-203. Genetic characterization and sequencing were done. This statement is not linked in anyway to prior discussion of heterogeneity in capsulation. Were any meaningful results found with the genetic characterization that can tie this statement to the prior discussion? Recommend adding something that makes this sentence meaningful.

This has been modified accordingly

  1. Lines 241-243. Longevity of spores. This sounds interesting but it would be even more so if the findings were included. What did they find? How many years or decades were spores still viable?

This has been modified accordingly

  1. Lines 272-276. This seems to suggest there was an interruption in B. anthracis research from the early 1970’s until 1986. Yet the example mentioned as evidence that some experiments were performed (ref 25) is from before this time period, 1966. This doesn’t fit the timeline. Was work on the vaccines continued until 1970’s but ‘research’ was interrupted before the 1970’s? Please clarify.

 We are thankful to the reviewer for spotting this inconsistency; We have added some details.

  1. Lines 327-328. Dependence of calmodulin for EF needs a reference.

This has been modified accordingly

  1. Lines 516-518. Place the parentheses explaining the acronym IRBA right after IRBA, then the medical research structure of the army at the end of the sentence.

This has been modified accordingly

  1. Lines 552-556. Interesting results. Discussion or thoughts on why adding toxin component to spores for vaccination gives better protection against non-toxigenic strain than spores alone?

This has now been addressed

  1. Line 629. Not familiar with the term ‘patrimony’ in this context. Is there an alternative to explain what was destroyed? Historical experimental collections?

We used the terminology suggested to us, this not being our expertise. As mentioned in the general appreciation, the ms "is low key" : there is globally some reticence in officially discussing such aspects. Clearly independant  research on this aspect could be highly valuable to fully appreciate the impact of this legislation.We preferred to remain diplomatic...

Comments on the Quality of English Language

This needs some work. It is recommended that a native English speaker with good writing skills review the draft for sentence structure and flow. There are just a few minor instances listed below. 

In a few places, 'as soon as' for work of an early date should be replaced with 'as early as'. This means 'as early in time as'. 'Soon' is generally used to describe something that might be done as soon as possible or in the near future.

Line 179. Use 'lose' instead of 'loose'.

Footnote #12. Vaccine development is either 1) 'subject to' or 2) 'confronted by' many hurdles.

Lines 689. It's probably better to say 'genetically manipulate' than 'manipulate genetically.

These points have been modified accordingly

Reviewer 2 Report

Comments and Suggestions for Authors

The review proposed by the authors is a very interisting historogical work on the B.anthracis. I have only few commentents.

1. line 49: check the sentence

2. line 64: please change cattle in " domestic and wild herbivorous animals"

3. line 251: check the sentence

4. line 498: please change "heroin users" in "drug users"

Author Response

Referee 2 :

The review proposed by the authors is a very interisting historogical work on the B.anthracis. I have only few commentents.

  1. line 49: check the sentence
  2. line 64: please change cattle in " domestic and wild herbivorous animals"
  3. line 251: check the sentence
  4. line 498: please change "heroin users" in "drug users"

 These points have been modified accordingly

Reviewer 3 Report

Comments and Suggestions for Authors

This review by anonymous author(s) describes the long history of research concerned with Bacillus anthracis and anthrax at Institut Pasteur in Paris (France). Celebrated and all but forgotten heroes and their scientific breakthroughs are described along with highlight pieces of their works. This informative compilation comes along with the grim situation of anthrax research at Institut Pasteur or France at large. Of course, the review is also a sentimental journey given the scope covering the golden ages of microbiology in general and the particularly seminal historic anthrax research at Institut Pasteur. Details of the report are witness of the sad irony of history when the author(s) state “A scientific collaboration was finally taking place between our two institutions [the Institut Pasteur and the Robert Koch-Institut] more than one hundred years after the fierce competition between Louis Pasteur and Robert Koch in the 1880's” in the context of anthrax research having ceased altogether at Institut Pasteur in the second decade of the XXI century. It is mind-boggling to imagine the scope of loss for continued research in terms of previously collected intellectual achievements (now buried in archives) and of biological resources (e.g. strain collection, historic specimens). On the brighter side, the author(s) present some valid ideas in their text for researcher of our time to continue on topics set forth at Institut Pasteur when it still flourished.

General comments:

1)   The manuscript is generally well written and mostly easy to read. There are only few (relatively) minor issues as outlined below (including grammar and punctuation, possibly best explained by the author(s) romance language background; for instance in line 153). These minor issues can be fixed at the stage of typesetting, though.

Specific minor comments:

2)   Line 12 (and elsewhere): please replace slang “lab” with “laboratory”.

3)   L 64: abbreviate Bacillus anthracis after its first occurrence in the text and check if it is always italicized (e.g., in L 142 and in the title too).

4)   L 97 (and elsewhere): what does AIP SVV stand for? Please explain

5)   L 133: please replace “traced as soon as…” with “traced back to…”.

6)   L139-140: please change to “…onwards; from 1934…until 1951, the date…”.

7)   L 164: better “…species to be vaccinated showed different susceptibilities…”.

8)   L 168: better “…prompting additional research…”.

9)   L 171: “bovines”

10)               L 190: better “…were related to…”.

11)               L 280: “tetanus”

12)               L 287: please replace “advances” with “progress”.

13)               L 351: please replace “hints” with “clues”.

14)               L482-: this paragraph is confusing. Why this jump back in time to the pre-antibiotic era when the topic of this section is the late XX century? Maybe it is just ambiguous writing?

15)               Footnote 10: “Their” refers to what?

16)               L 500-501: “antibiotic therapy”

17)               L 502: better “..may have been…”.

18)               L 504: better “..such (nefarious) uses”.

19)               L 510: better replace “It” with “Their”.

20)               L 506/516 (and elsewhere). Please replace “actors” with “authorities”.

21)               L 527: “Sterne strain”

22)               L 531: “…raised on the actual efficiency of vaccines in real…”.

23)               L 533: “…therapeutics in general, and more specifically for anthrax…”.

24)               L 563: “…did not afford…”.

25)               Footnote 12: “…confronted by many…”. […] ”…while safe, to be managed …”.

26)               Footnote 18 (L 567): “…OF1 mice. Results were synthesised from four independent…[…]; (L572) “Results were synthesised from five independent…”.

Comments on the Quality of English Language

There are only few (relatively) minor language issues mostly related to grammar (e.g., word order) and punctuation. These minor issues can be fixed at the stage of typesetting, though.

Author Response

Referee 3 :

This review by anonymous author(s) describes the long history of research concerned with Bacillus anthracis and anthrax at Institut Pasteur in Paris (France). Celebrated and all but forgotten heroes and their scientific breakthroughs are described along with highlight pieces of their works. This informative compilation comes along with the grim situation of anthrax research at Institut Pasteur or France at large. Of course, the review is also a sentimental journey given the scope covering the golden ages of microbiology in general and the particularly seminal historic anthrax research at Institut Pasteur. Details of the report are witness of the sad irony of history when the author(s) state “A scientific collaboration was finally taking place between our two institutions [the Institut Pasteur and the Robert Koch-Institut] more than one hundred years after the fierce competition between Louis Pasteur and Robert Koch in the 1880's” in the context of anthrax research having ceased altogether at Institut Pasteur in the second decade of the XXI century. It is mind-boggling to imagine the scope of loss for continued research in terms of previously collected intellectual achievements (now buried in archives) and of biological resources (e.g. strain collection, historic specimens). On the brighter side, the author(s) present some valid ideas in their text for researcher of our time to continue on topics set forth at Institut Pasteur when it still flourished.

 General comments:

1)   The manuscript is generally well written and mostly easy to read. There are only few (relatively) minor issues as outlined below (including grammar and punctuation, possibly best explained by the author(s) romance language background; for instance in line 153). These minor issues can be fixed at the stage of typesetting, though.

 Specific minor comments:

All these points have been addressed accordingly. Some precisions are added more specifically

2)   Line 12 (and elsewhere): please replace slang “lab” with “laboratory”.

We left "lab books", though it could be replaced by "laboratory notebooks". We will conform to the edito'sr recommendation

3)   L 64: abbreviate Bacillus anthracis after its first occurrence in the text and check if it is always italicized (e.g., in L 142 and in the title too).

4)   L 97 (and elsewhere): what does AIP SVV stand for? Please explain

This is specified in the abbreviation list ("Archives de l'Instiut Pasteur" "Service des Vaccins Vétérinaires"), that was transferred to the end of the ms.

5)   L 133: please replace “traced as soon as…” with “traced back to…”.

6)   L139-140: please change to “…onwards; from 1934…until 1951, the date…”.

7)   L 164: better “…species to be vaccinated showed different susceptibilities…”.

8)   L 168: better “…prompting additional research…”.

9)   L 171: “bovines”

10)               L 190: better “…were related to…”.

 We are not sure if the modification we made was at the correct location

11)               L 280: “tetanus”

12)               L 287: please replace “advances” with “progress”.

13)               L 351: please replace “hints” with “clues”.

14)               L482-: this paragraph is confusing. Why this jump back in time to the pre-antibiotic era when the topic of this section is the late XX century? Maybe it is just ambiguous writing?

We reworked this part to emphasize the veterinary part of anthrax control, before turning to the human part

15)               Footnote 10: “Their” refers to what?

16)               L 500-501: “antibiotic therapy”

17)               L 502: better “..may have been…”.

18)               L 504: better “..such (nefarious) uses”.

19)               L 510: better replace “It” with “Their”.

20)               L 506/516 (and elsewhere). Please replace “actors” with “authorities”.

21)               L 527: “Sterne strain”

22)               L 531: “…raised on the actual efficiency of vaccines in real…”.

23)               L 533: “…therapeutics in general, and more specifically for anthrax…”.

24)               L 563: “…did not afford…”.

25)               Footnote 12: “…confronted by many…”. […] ”…while safe, to be managed …”.

26)               Footnote 18 (L 567): “…OF1 mice. Results were synthesised from four independent…[…]; (L572) “Results were synthesised from five independent…”.